# Performance analysis of deep neural network based on channel pruning

Junfeng Chen*
*College of Artificial Intelligence and Automation, Hohai University*
*Jiangsu Key Laboratory of Power Transmission & Distribution Equipment Technology, Hohai University*
Changzhou, China
chen-1997@163.com

Na Li
*College of Artificial Intelligence and Automation, Hohai University*
Changzhou, China
lina1783@163.com

Ziyang Weng
*College of Artificial Intelligence and Automation, Hohai University*
Changzhou, China
wengtianx@gmail.com

Jingjing Du
*College of Artificial Intelligence and Automation, Hohai University*
Changzhou, China
hzdujing@163.com

*Abstract*—**Model compression technology, a crucial aspect of neural network models, offers a range of benefits. It reduces the number of parameters and computational load, thereby shrinking the model size, enhancing inference speed, decreasing memory usage, and saving power. This article delves into the research of model compression technology for neural network models, focusing on channel pruning algorithms and model compression methods based on the Batch Normalization (BN) layer. The goal is to reduce the number of model parameters and computational load, leading to a smaller model size, faster inference speed, reduced memory usage, and saved power. The article applies sparse regularization to the scaling factors of the BN layer, serving as the basis for determining channel importance and reducing model complexity. It then presents experimental comparisons on VGGNet-16, ResNet-164, and DenseNet-40 neural network models, including standard training, sparse regularization, and pruning fine-tuning training results. The experiments reveal that the pruned networks achieve comparable or even higher accuracy than the original networks, underscoring the importance of the research in model compression technology.**

*Keywords—channel pruning, convolution neural networks, model compression*

## I. INTRODUCTION

Deep Neural Networks (DNN) have unique architecture and characteristics to adapt to different tasks and data sets, but as the depth and width of the model increase, the computational resources required for its training and reasoning increase dramatically. For example, the number of parameters in Alex Net [1] reaches 60M, and even the computation amount in some image training tasks reaches hundreds of millions of floating points. Although the development of Graphics Processing Unit (GPU) accelerates the training of models, in actual application scenarios, network bandwidth limitations make edge computing devices a necessary choice. Embedded platforms have small memory and limited computing power, so it is necessary to compress the neural network model to adapt to resource constraints [2]. To solve these problems, researchers have proposed compression methods such as low-rank decomposition, weight quantization, knowledge distillation, and model pruning [3]. However, these methods can only solve some of the above problems, and some methods still need to rely on specific software and hardware acceleration.

One practical approach to reduce resource consumption in large Convolutional Neural Network (CNN) is through network sparsity. This method introduces sparsity at different levels, significantly compressing model size and speeding up inference. Anwar [4] pioneered the inclusion of sparsity in feature maps, kernels, and their internals within pruning strategies, achieving substantial model compression. Subsequent research by Han [5] explored balancing network sparsity with accuracy, proposing methods to maintain high precision while promoting structured sparsity. Lebedev [6] also applied group pruning strategies to convolutional kernels, enhancing computational efficiency through sparse regularization. Wen [7] developed structured sparse learning algorithms that regulate filter and channel structures, effectively reducing model size and computational burden. These studies demonstrate that sparsity strategies can significantly decrease computing and storage costs while maintaining model performance.

This study explores a practical and effective method for channel pruning, addressing challenges in deploying large CNN under limited resources. The approach involves sparse regularization of Batch Normalization (BN) layer scaling factors as criteria for channel importance assessment without requiring modifications to the existing model framework. By applying L1 regularization, the scaling factors of BN layers tend towards zero, automatically identifying unimportant channels and significantly reducing model complexity. Compared to the original network, the pruned network achieves much compactness regarding model size, runtime memory, and computational requirements. Iterating this process multiple times yields a multi-channel model compression approach, making the network more streamlined and practical for real-world applications.

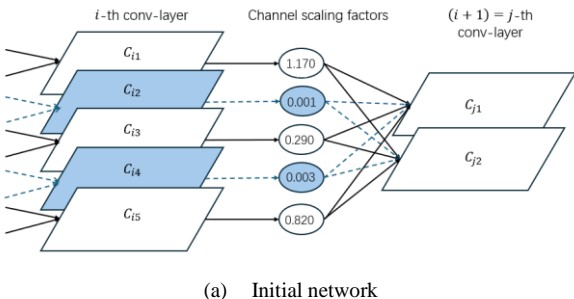

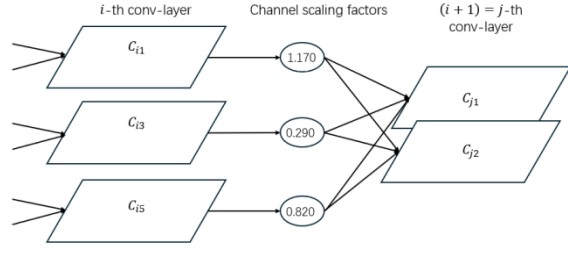

| (a)   Initial network | (b)   Initial network |

Fig. 1.  Comparison of deep neural network structures before and after pruning operation

## II. CHANNEL PRUNING

Channel pruning is a structured pruning technique that evaluates the contribution and importance of each channel within a neural network, identifying and removing non-critical channels that have minimal impact on model performance. This method significantly reduces model complexity and computational load, enhancing inference speed and efficiency without sacrificing accuracy. It makes the model more suitable for resource-constrained environments or real-time applications. Implementing channel pruning involves three key steps: evaluating channel importance, selecting channels for pruning, and executing the operation. This process helps optimize model performance and improve the efficiency of computational resource utilization.

This paper evaluates channel importance by using the values of trainable scaling factors on each channel for pruning. Specifically, channels corresponding to scaling factors close to zero are considered unimportant and pruned, treating channel pruning as a sparsity optimization problem. BN layers are commonly used in neural networks to accelerate training and enhance model generalization. The scale factors within BN layers reflect the importance of each channel in the model. By introducing regularization and adjusting the values of channel scale factors, unimportant channels have their scale factors approach zero. The specific values of these scale factors are then used to measure channel importance, automatically removing channels with relatively lower importance. This approach compresses the model while maintaining accuracy, resulting in a compact neural network, as shown in Fig. 1.

## III. CHANNEL PRUNING-BASED DEEP NEURAL NETWORK MODEL COMPRESSION METHOD

### A. Importance evaluation and L1 sparse training

The core idea of channel pruning is to remove redundant channels to simplify the model. Because channel pruning involves pruning and thinning parts of the network structure rather than individual weights, it does not require specialized libraries to achieve inference speedup and runtime memory savings [8]. The mathematical expression of channel pruning is represented as:

$$\arg\min_{\beta,W} \frac{1}{2N}\|Y - \sum_{i=1}^{c}\beta_i X_i W_i\|_F^2,$$

$$s.t. \ \|\beta\|_0 \leq c', 0 \leq c' \leq c \tag{1}$$

where $c$ represents the number of channels, $i$ is the index of the number of channels, $c'$ represents the number of channels, $X_i$ and $W_i$ correspond to each channel of the input feature map and convolution kernel respectively, and $\beta_i$ denotes channel coefficients.

If $\beta$ equals 0, the corresponding channel will be removed. The ordinary least square method can be used to solve the pruning problem without affecting the accuracy of the model. This paper aims to compress the input feature map channels from $c$ to $c'$ while minimizing the reconstruction error as much as possible. Because $\beta$ does not participate in the weight parameter update process in the above formula, its value can only be 0 or 1, complicating the optimization process.

For channel pruning, finding an appropriate criterion to assess channel importance is necessary to ensure effective model pruning. In image recognition tasks, BN layers are included in CNN to prevent gradient explosion, accelerate model convergence, and enhance generalization performance [11]. BN layers normalize data using trainable scaling factors $\gamma$ and offset coefficients $\beta$, allowing them to learn the feature distribution of convolution layer outputs. Consider a convolution layer's input composed of $b$ samples; the layer's output features form a 4th-order tensor $x \in R^{b \times c \times H \times W}$, where $H$ and $W$ represent the height and width of the feature map, respectively, and $c$ denotes the channel dimension of the feature map. The BN layer first computes the mean $\mu$ and variance $\sigma^2$ for each channel in the feature map $x$. For any channel $v \in \{1,2,\cdots;c\}$, formulas (2) and (3) are satisfied.

$$\mu_V = \frac{1}{b \times h \times w}\sum_{i=1}^{b}\sum_{j=1}^{h}\sum_{k=1}^{w} x_{i,v,j,k} \tag{2}$$

$$\sigma_V^2 = \frac{1}{b \times h \times w}\sum_{i=1}^{b}\sum_{j=1}^{h}\sum_{k=1}^{w}(x_{i,v,j,k} - \mu_V)^2 \tag{3}$$

Normalize the feature map $X$ after the convolution operation. Where $X_{(v)}$ is the tensor of feature maps on batch data with channel index $v$.

$$\hat{Z}_{(v)} = \frac{X_{(v)} - \mu_v}{\sqrt{\sigma_v^2 + \varepsilon}} \tag{4}$$

The batch normalization layer plays a crucial role in stabilizing the back propagation process during model training.

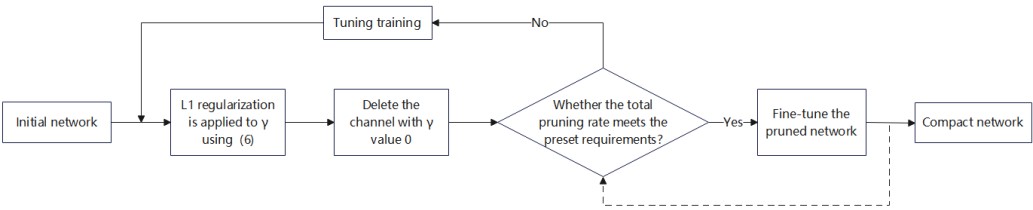

Fig. 2.   Flow chart of iterative pruning strategy

It achieves this by introducing scaling factors $\gamma$ and offset coefficients $\beta$. These two sets of parameters are responsible for the stretching and shifting operations performed by the BN layer on its results. The output of the feature maps after batch normalization processing is represented as:

$$Z_{out} = \gamma_v \hat{Z}_v + \beta_v \tag{5}$$

When the scaling factor $\gamma_v$ in the equation approaches zero, the output of the BN layer tends to $\beta_v$ for potential biases introduced during the normalization operation. In neural networks, each convolutional layer is typically followed by a BN layer, where the scaling factors $\gamma$ in the BN layer correspond one-to-one with the channels of each convolutional layer.

Thus, the scaling factor's magnitude can quantify each channel's contribution to the model's performance. If the scaling factor $\gamma$ of a channel approaches zero, it indicates that the feature maps generated by that channel contribute minimally to the model's performance, making the channel redundant. By removing these redundant channels, the parameter count of the network can be reduced without altering the original network's feature extraction capability.

To guide the network in producing sparse scaling factors $\gamma$ during training, we apply L1 regularization to $\gamma$. This regularization technique encourages the scaling factors $\gamma$ to converge towards smaller values, thereby promoting pruning. The effect of L1 regularization is to make some values of the scaling factor distributed near 0, effectively making some channels less important in the network's performance. Define an objective function as:

$$L = \sum_{(x,y)} l(f(x,W),y) + \lambda \sum_{\gamma \in \Gamma} g(\gamma) \tag{6}$$

where $(x,y)$ represents the training data and corresponding target values, and $W$ denotes the trainable weight parameters in the network.

The first term is the CNN's training loss function, which measures the difference between the predictions made by the network and the actual target values. The second term, $g(\gamma)$, is a sparsity penalty term on the scaling factor designed to encourage these factors to converge toward smaller values, thereby promoting pruning. The $\lambda$ is a balancing factor to adjust the relative importance between training loss and sparsity penalty [9]. This paper adopts $g(s)=|s|$, utilizing L1 regularization.

We use the Stochastic Gradient Descent (SGD) method to calculate the gradient of the loss function $L$ concerning the scaling factor, as shown below.

$$\nabla_\gamma L = \nabla_W l \nabla_\gamma W + \lambda sgn(\gamma) \tag{7}$$

During training, the network learns the $\gamma$ value of each channel. After the training, the BN layer's $\gamma$ value is statistically sorted. Based on the set pruning threshold, the network then creates a mask of the same dimension as the $\gamma$ value to indicate which channels need to be pruned. If a $\gamma$ value is less than the threshold, the corresponding mask value is set to 0, and the channel can be deleted. Otherwise, the value is 1, and the channel is reserved. When all the channels with a mask value of 0 are pruned from the network, we have a compact neural network model. Finally, we also need to fine-tune the pruning model to compensate for the loss of accuracy.

### B. Iterative pruning strategy

The single pruning learning method includes sparsity regularization training, pruning, and fine-tuning. However, to avoid the significant degradation of network performance, we adopt the iterative pruning method. This method breaks down the pruning process into multi-stage processes, ensuring a systematic approach to network optimization. We delete one section of the network at a time under the overall pruning rate target. We update the loss function and proceed to the tuning process for the network to reduce the cumulative error caused by deleting the channel. Then, the importance of the remaining channels is calculated using (6), and the next pruning channels are sorted. When the number of iterative pruning parameters to the model meets the preset requirements, the pruning is stopped. Finally, it's time to fine-tune. The entire training process is applied several times to learn a more compact network model. The iterative pruning strategy is shown in Fig. 2.

During channel pruning, channels are ranked based on $\gamma$, which reflects their importance in the model. A portion of less important channels is then pruned according to a set iterative pruning rate. This iterative pruning approach allows adjustments based on the previous round's results, ensuring a balanced and optimized pruning effect and speed. In Fig. 2, the dashed line indicates that if fine-tuning results are unsatisfactory, the overall pruning rate is reduced.

### C. Channel selection module

Scaling factors from BN layers as a criterion for channel importance in pruning methods can be directly applied to conventional CNN architectures, such as AlexNet [1] and VGGNet.

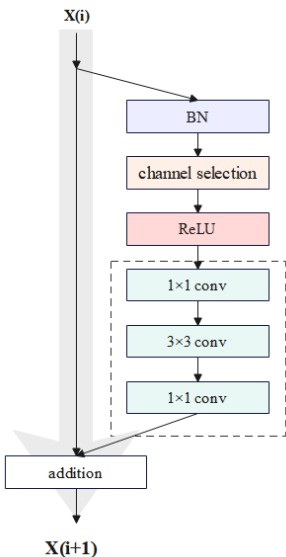

Fig. 3. Structure Diagram of ResNet-164 with CS Module

However, it does not perform well for networks like ResNet [10] and DenseNet [11], which feature skip connections and pre-activation designs, where the output of each layer serves as input to multiple subsequent layers. Typically, a BN layer is placed before convolutional layers between these layers. In such cases, sparsity is achieved at the input side of the layer, meaning the layer selectively uses a subset of channels it receives.

To achieve savings in parameters and computations, a channel selection (CS) module is added after the BN layer instead of directly pruning convolutional layers within each layer[9]. Taking ResNet-164 [10] as an example, ResNet-164 adopts a pre-activation structure, where BN and activation functions (such as ReLU) are applied before convolution operations in each residual unit. This design helps mitigate the vanishing gradient problem, enhancing model training efficiency and convergence speed. The network model with the CS module added to each residual unit of this architecture is depicted in Fig. 3. The dotted box represents bottleneck structure.

Therefore, rather than directly pruning convolutional layers, introducing a CS module after the BN layer allows channel selection based on the parameter. This approach is more flexible in implementation, enabling savings in parameters and computations while maintaining network structural stability. By leveraging the CS module, the subset of input channels received by each layer can be dynamically adjusted by parameters during training to achieve sparsity and performance optimization.

## IV. EXPERIMENTAL VALIDATION AND ANALYSIS

### A. Experimental settings

This study conducted experiments on three network models: VGGNet, ResNet-164, and DenseNet-40, using two CIFAR datasets for training and testing. The experiments were conducted with 160 epochs, optimizing the loss function of the network models using the SGD algorithm. A weight decay coefficient $\lambda$ of $10^{-4}$ and a Nesterov momentum [12] of 0.9 without dampening were chosen. The batch size was 64, and the initial learning rate was set to 0.1. During training, the learning rate was adjusted to 0.01 at epoch 80 and 0.001 at epoch 120 to facilitate learning rate updates.

This study trained the models in normal conditions using standard training procedures without compression or optimization. These results serve as the baseline for comparison with other conditions. The scaling factor for channels typically defaults to 1. However, the experiments set it to 0.5 to better suit the current task or model architecture characteristics, thereby potentially improving training accuracy. When pruning the channels of models trained through sparsity, the choice of threshold values directly impacts the model's overall performance. Therefore, determining an appropriate scaling factor pruning threshold is crucial.

When the pruning threshold is too large, it results in minimal or negligible pruning. In contrast, excessively small thresholds may lead to over-pruning, removing too many weights or structures from the model. Therefore, when the pruning ratio exceeds a certain threshold, both pruning and fine-tuning may cause a decline in classification performance. However, fine-tuning can often compensate for accuracy losses due to excessive pruning. Hence, during the experimental pruning process, this study tested the model's accuracy under different pruning thresholds to determine the optimal threshold. During fine-tuning, epochs are set to 160, and the model's accuracy and error values are recorded after each epoch to determine the optimal epoch. Upon completion of training, the accuracy and parameter count of the models trained under normal conditions and after pruning are compared.

### B. Experimental results and analysis

In this section, three network models were trained on the CIFAR dataset.

TABLE I. RESULTS ON CIFAR-10

| Models | CIFAR-10 | Baseline | Sparsity | Prune | Fine-tune |
|---|---|---|---|---|---|
| VGG Net (70%) | Accuracy (%) | 91.81 | 91.70 | 32.54 | 91.78 |
| | Parameters (M) | 20.04 | 20.04 | 2.25 | 2.25 |
| ResNet-164 (40%) | Accuracy (%) | 88.83 | 88.76 | 12.32 | 88.05 |
| | Parameters (M) | 1.71 | 1.71 | 1.45 | 1.45 |
| DenseNet-40 (40%) | Accuracy (%) | 90.11 | 90.17 | 25.69 | 90.32 |
| | Parameters (M) | 1.10 | 1.10 | 0.69 | 0.69 |

TABLE II. RESULTS ON CIFAR-100

| Models | CIFAR-10 | Baseline | Sparsity | Prune | Fine-tune |
|---|---|---|---|---|---|
| VGG Net (50%) | Accuracy (%) | 71.12 | 70.85 | 5.31 | 71.32 |
| | Parameters (M) | 20.04 | 20.04 | 4.93 | 4.93 |
| ResNet-164 (40%) | Accuracy (%) | 66.07 | 66.13 | 48.00 | 67.36 |
| | Parameters (M) | 1.71 | 1.71 | 1.49 | 1.49 |
| DenseNet-40 (40%) | Accuracy (%) | 70.33 | 68.88 | 60.67 | 70.76 |
| | Parameters (M) | 1.10 | 1.10 | 0.71 | 0.71 |

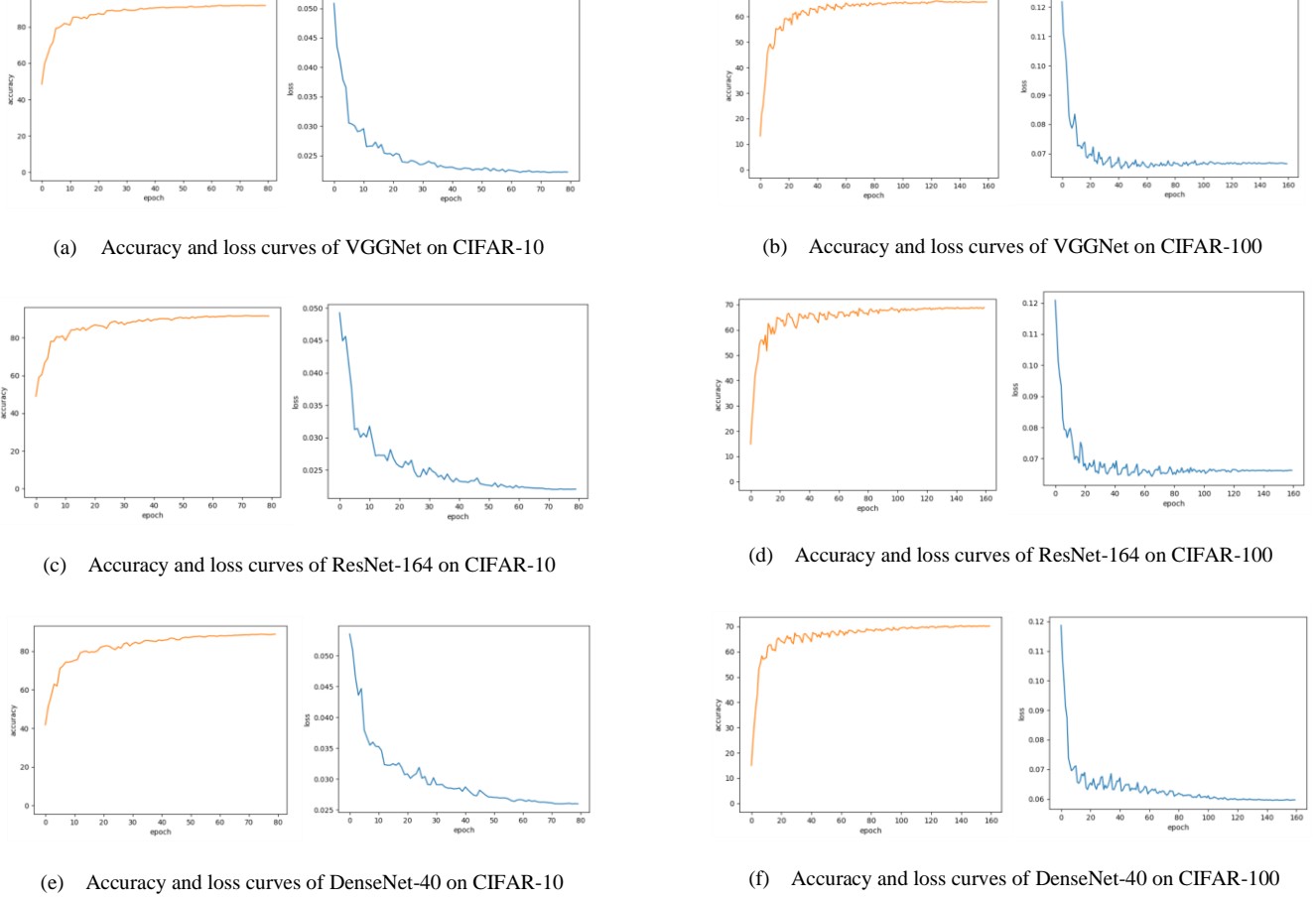

(a) Accuracy and loss curves of VGGNet on CIFAR-10

(b) Accuracy and loss curves of VGGNet on CIFAR-100

(c) Accuracy and loss curves of ResNet-164 on CIFAR-10

(d) Accuracy and loss curves of ResNet-164 on CIFAR-100

(e) Accuracy and loss curves of DenseNet-40 on CIFAR-10

(f) Accuracy and loss curves of DenseNet-40 on CIFAR-100

Fig. 4. Accuracy curves of three network models on CIFAR

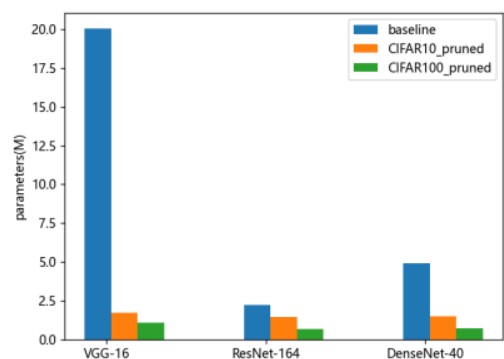

Fig. 5. Comparison of the number of parameters of the three deep learning models before and after the pruning operation

TABLE I. shows that after fine-tuning the VGGNet-16 model trained on CIFAR-10, the model achieved comparable accuracy to normal training while reducing the parameter count by 17.79 million. Further comparison of the loss values between normal training and 70% pruned plus fine-tuned VGGNet-16 models indicates that the pruned model's loss remains comparable to normal training, suggesting that pruning did not significantly degrade model accuracy but effectively reduced parameter count.

Additionally, results from fine-tuning ResNet-164 and DenseNet-40 models show a slight decrease in accuracy by 0.78% and a parameter reduction of 0.26 million for ResNet-164. In contrast, DenseNet-40 exhibited a slight accuracy improvement of 0.21% with a parameter reduction of 0.41 million. These findings demonstrate that pruning can effectively enhance model performance by mitigating the impact of unimportant channels on accuracy, thereby achieving parameter reduction objectives.

Fig. 1 presents the training results of three network models on the CIFAR-100 dataset. After fine-tuning, VGGNet-16 experienced a slight accuracy decrease of 0.2% compared to regular training, while the parameter count dropped significantly from 20.04 million to 4.93 million. The pruned VGG model maintained its accuracy and significantly reduced its parameter count. For the ResNet-164 model on the CIFAR-100 dataset, fine-tuning resulted in an accuracy improvement of 0.29% compared to regular training, along with a reduction of 0.22 million parameters. Similarly, the DenseNet-40 model showed an accuracy improvement of 0.43% and a parameter reduction of 0.39 million after fine-tuning. These results demonstrate that fine-tuning is an effective optimization method that can reduce model parameter count while maintaining or improving model accuracy.

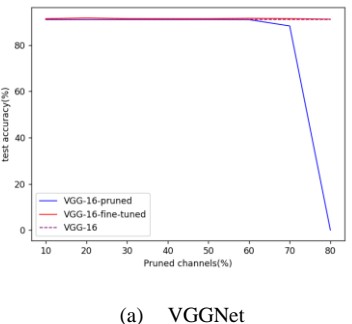
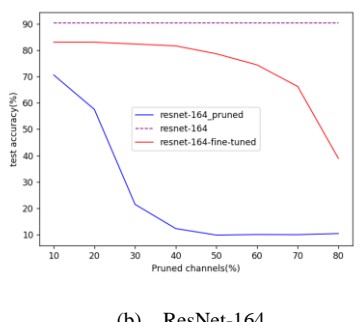
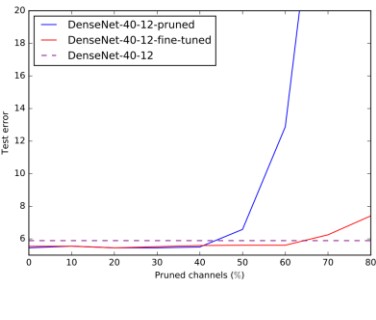

| (a) VGGNet | (b) ResNet-164 | (c) DenseNet-40 |

Fig. 6. Accuracy curves of three network models on CIFAR

Fig. 4 shows the changes of accuracy and loss of the three network models in the training process. To find the optimal pruning threshold, we conducted comparative experiments with different pruning thresholds on three network models. Various pruning thresholds were selected to prune the models, followed by fine-tuning. The test errors were recorded over 30 epochs on the CIFAR-10 dataset. The results are shown in 0. For the VGGNet-16 network, the optimal pruning ratio is 70%. Beyond this threshold, the performance of the pruned model significantly deteriorates. Similarly, for the ResNet-164 network, the optimal pruning ratio is 40%. In the case of the DenseNet-40 network, pruning ratios exceeding 40% show an increase in error post-pruning. However, subsequent fine-tuning effectively compensates for this accuracy loss, with the best performance observed at the 40% pruning ratio. However, when the pruning ratio is further increased to above 70%, the test error of the fine-tuned model gradually becomes worse than that of the unpruned baseline model.

Fig. 5 compares parameter counts before and after pruning for the three models. After pruning, the parameter counts of all three models decreased. Particularly for VGGNet-16, the parameter counts significantly reduced after pruning, indicating that removing unimportant parameters from the network can greatly simplify the model complexity while maintaining accuracy. This demonstrates that pruning can effectively reduce model complexity without compromising accuracy.

## V. RESULTS

The channel pruning strategy adopted in this paper can significantly reduce the parameter count of CNNs while ensuring model performance remains intact. By applying L1 regularization to the scaling factors of Batch Normalization layers, the method effectively selects unimportant channels. This allows the network to automatically prune non-essential parameters during training, reducing both the number of parameters and the computational load. The pruning approach not only decreases model complexity but also maintains or even enhances performance in image classification tasks. Experimental validation across multiple image classification datasets and various network models shows that three pruned network models achieve the same or even higher accuracy compared to the baseline model, while the number of parameters is significantly reduced after pruning. These results underscore

the method's generality and robustness across different model architectures and training setups, applicable to various convolutional neural network models. Therefore, the channel pruning method proposed in this paper reduces computational complexity and preserves model generalization ability and classification accuracy.

## ACKNOWLEDGMENT

This work is partly supported by the Jiangsu Key Laboratory of Power Transmission & Distribution Equipment Technology (grant number 2023JSSPD07, 2022JSSPD05) and the Key Research and Development Plan of Jiangsu Province (grant number BE20219042).

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
