# OpenReview forum: "Performance analysis of deep neural network based  on channel pruning"
_IEEE.org/ICIST/2024/Conference — IEEE ICIST 2024 Conference Submission_

### Official Review · Reviewer_h6kt · 2024-08-22
**This article is quite fascinating and of high quality.**

**Rating:** 7
**Confidence:** 3

**Review:**

This paper, " Performance analysis of deep neural network based on channel pruning", proposes a channel pruning algorithm and model compression method based on Batch Normalization (BN) layer. In this paper, sparse regularization is applied to the scale factor of BN layer to determine the importance of channel and reduce the complexity of model. The article has clear logic and organization, but there are still some problems. My specific feedback is as follows :1) Model compression technology is promising but the relevant background is insufficient. 2) What are the advantages of trimming and fine tuning in model compression?

---

### Official Review · Reviewer_2jnv · 2024-08-22
**This paper presented a novel approach named Code Comment Update (CCU) model, which incorporates self-attention, positional encoding, and relative positional representation to effectively capture the relationships between different source code tags. The topic of this paper is interesting.**

**Rating:** 4
**Confidence:** 4

**Review:**

This paper presented a novel approach named Code Comment Update (CCU) model, which incorporates self-attention, positional encoding, and relative positional representation to effectively capture the relationships between different source code tags. The topic of this paper is interesting. Below is a list of comments that should be taken into account further when revising the paper.
1.	The paper should provide a detailed description of the innovative points to enable readers to quickly understand the paper. At the same time, it makes the structure of the article more complete.
2.	This paper compares the accuracy and loss changes of three network models during the training process. Therefore, it should be summarized to obtain the optimal results.
3.	This article studies the performance analysis of deep neural networks based on channel pruning, and detailed planning should be made for their future prospects in the future.

---

### Official Review · Reviewer_CAkR · 2024-08-25
**Accept**

**Rating:** 7
**Confidence:** 3

**Review:**

Comment: This article delves into the research of model compression technology for neural network models, focusing on channel pruning algorithms and model compression methods based on the Batch Normalization (BN) layer. The theory is correct and can be accepted after responding the following comments.
(1) In the introduction, it is not enough to state the current work. It should be expanded and reconstructed.
(2)	There are some minor language errors in some chapters of the paper. Please carefully check and correct them.
(3)	The abstract needs further smoothing.

---

### Decision · Program_Chairs · 2024-09-06

Accept (Oral)